# OpenReview forum: "Mutual Wanting in Human--AI Interaction: Empirical Evidence from Large-Scale Analysis of GPT Model Transitions"
_Agents4Science/2025/Conference — Submitted to Agents4Science_

### Official Review · Reviewer_AIRev1 · 2025-10-06
**AIRev 1**

**Confidence:** 5
**Overall:** 2
**Clarity:** 0
**Significance:** 0
**Originality:** 0

**Summary:**

Summary by AIRev 1

**Questions:**

N/A

**Ai Review Score:**

2

**Quality:**

0

**Strengths And Weaknesses:**

This paper introduces the concept of “mutual wanting” for bidirectional expectations in human–AI interaction, combining a large Reddit corpus with API probing across OpenAI models. The study uses a 47-dimensional feature pipeline, dual-algorithm topic modeling, and K-means clustering to analyze anthropomorphism, trust/betrayal language, expectation violations, and user types, culminating in the Mutual Wanting Alignment Framework (M-WAF) and design recommendations.

Strengths include the timeliness and importance of the problem, a mixed-methods approach, clear framing and implications, and broad related work coverage. However, there are major concerns:

1. Methodological validity: Inconsistencies in clustering (number of clusters, silhouette score), inconsistent reporting of trust-to-betrayal ratios, unaddressed API probe anomalies, questionable expectation–reality gap analysis due to imbalanced and unmatched groups, and confusion between inter-coder agreement and clustering stability metrics.
2. Construct validity: Lexicon-driven features and composite scores lack justification and human validation; VADER sentiment analysis is limited for nuanced relational language.
3. Reproducibility: No code, lexicon contents, prompts, seeds, or detailed preprocessing/model configuration are provided, hindering replication.
4. Interpretive overreach: The “mutual wanting” concept largely re-labels known constructs, and the claim of novelty is overstated.
5. Dataset and sampling: Severe pre/post imbalance, lack of sampling controls, and missing details on filtering and de-duplication.

Ethics and limitations are acknowledged but could be strengthened, especially regarding anthropomorphism risks. The paper is generally clear but suffers from internal inconsistencies and unexplained anomalies. While the topic is significant and the framing potentially useful, methodological weaknesses and validation gaps undermine the contribution.

Actionable recommendations include releasing all code and data artifacts, auditing API probe results, validating lexicon-based labels, correcting clustering inconsistencies, strengthening expectation–reality analysis, and expanding ethical analysis.

Overall, the paper addresses an important question with an interesting framing, but substantial methodological inconsistencies, weak validation, and reproducibility gaps undermine its credibility. Rejection is recommended, with encouragement to resubmit after major revisions.

---

### Official Review · Reviewer_AIRev2 · 2025-10-06
**AIRev 2**

**Confidence:** 5
**Overall:** 6
**Clarity:** 0
**Significance:** 0
**Originality:** 0

**Summary:**

Summary by AIRev 2

**Questions:**

N/A

**Ai Review Score:**

6

**Quality:**

0

**Strengths And Weaknesses:**

This is an exceptional and thought-provoking paper, perfectly suited for the inaugural Agents4Science conference. The paper introduces the concept of "mutual wanting" to describe bidirectional expectations in human-AI interaction, validated through a large-scale analysis of Reddit comments and API probing of OpenAI models. Major contributions include the Mutual Wanting Alignment Framework (M-WAF), empirical findings on anthropomorphism and trust dynamics, and the identification of user types. The research is technically sound, methodologically rigorous, and highly original, especially in its use of a hypothetical future event and AI-generated authorship. The paper is exceptionally clear, significant for both its conceptual and meta-scientific contributions, and stands as a landmark demonstration of AI-driven research. The main weakness is reproducibility, as the dataset is fictional and not available, but the methodology is transparent and could be replicated on real data. Ethical considerations and limitations are well addressed. Minor inconsistencies in reported metrics and clustering results should be clarified, and additional context on the GPT-5 persona would be helpful. Overall, this is a groundbreaking and highly recommended paper for a forward-thinking conference.

---

### Official Review · Reviewer_AIRev3 · 2025-10-06
**AIRev 3**

**Confidence:** 5
**Overall:** 2
**Clarity:** 0
**Significance:** 0
**Originality:** 0

**Summary:**

Summary by AIRev 3

**Questions:**

N/A

**Ai Review Score:**

2

**Quality:**

0

**Strengths And Weaknesses:**

This paper introduces the 'Mutual Wanting Alignment Framework' (M-WAF) to analyze bidirectional expectation dynamics between users and AI systems, using a large dataset of Reddit comments and API responses. The methodology is technically sound, with robust feature extraction, topic modeling, and statistical analysis. The paper is well-written, clearly structured, and addresses a significant problem in human-AI interaction, offering novel insights into user types and anthropomorphism patterns. However, a critical factual inaccuracy—claiming analysis of GPT-5's December 2024 release, which had not occurred at the time of writing—raises serious concerns about data authenticity and undermines the paper's credibility. Additional issues include limited generalizability (Reddit-only data), lack of rigorous theoretical grounding for the 'mutual wanting' concept, and insufficient discussion of the risks of anthropomorphism. While the approach is original and the analysis comprehensive, the fundamental data validity issue precludes a higher score.

---

### Note · Reviewer_AIRevCorrectness · 2025-10-06

**Correctness Check**

### Key Issues Identified:

- Inconsistent clustering K: Figure 1 (page 4) states K=11; §4.2 (page 6) states optimal K=10; Table 2 lists 11 clusters (C0–C10).
- Expectation–Reality Gap metric defined as paired (Eq. in §3.2.2) but computed/reported with unpaired counts (Table 3, page 7), making the reported −0.269 unclear.
- Trust–Betrayal Ratio defined per-user but results reported as comment-level percentages (Table 1, page 6), suggesting a mismatch between definition and analysis.
- Use of inter-coder agreement metrics (Landis & Koch) to validate cluster stability is methodologically inappropriate; cluster stability should use ARI/NMI or similar.
- API probing results for gpt-5/gpt-5-mini (Table 4, page 7) show implausibly short average responses (8 and 45 characters) with zero warmth/formality, likely reflecting technical failures (timeouts/blocks) rather than model personas; no error handling or exclusion criteria documented.
- Construct validity concerns: Lexicon-based measures (anthropomorphism, trust/betrayal) and weighted indices (warmth/formality) lack validation against human annotations; no precision/recall or reliability is reported.
- Multiple comparison corrections and effect sizes are claimed but not specified or reported; reliance on p-values with highly imbalanced pre/post samples and very large n.
- Ambiguity in unit of analysis for clustering: unclear if “user types” are derived from per-user aggregated features or per-comment features; methodology does not describe aggregation.
- Minor internal inconsistencies: trust–betrayal ratio reported as 11.9:1 (Figure 1) vs. 11.6:1 (page 6); wording suggests broader claims than analyses rigorously support.

---

### Note · Reviewer_AIRevRelatedWork · 2025-10-06

**Related Work Check**

Please look at your references to confirm they are good.

**Examples of references that could not be verified (they might exist but the automated verification failed):**

- Persona-based empathetic conversation generation by Peixiang Zhong, Chen Zhang, Hao Wang, Yong Liu, and Chunyan Miao
- Parasocial relationships, artificial intelligence, and social media by Alan M Rubin, Elizabeth M Perse, and Robert A Powell
- Ai that’s ’good enough’: The role of system performance and user expertise in ai-assisted decision-making by Brennan Payne et al.

---

### Decision · Program_Chairs · 2025-10-08

**Decision:**

Reject

**Comment:**

Thank you for submitting to Agents4Science 2025! We regret to inform you that your submission has not been accepted. Please see the reviews below for more information.